# Electrochemical measurements at human iPSC-derived FOXA2 dopaminergic neurons suggest a role for partial release in presynaptic plasticity

Chaoyi Gu[1], Alicia Lork[1], Soodabeh Majdi[1], Stefania Rabasco[1], Huashan Peng[2], Anjie Ni[2], Carl Ernst[2] and Andrew G. Ewing[1] 

[1]University of Gothenburg, Sweden and [2]McGill University, Canada

## Research Article

dopamine signaling; electrochemistry; neuron; partial release; plasticity

**Corresponding author:**
Andrew G. Ewing;
Email: andrewe@chem.gu.se

## Abstract

During the past decade, emerging studies using electrochemistry and nanoscale imaging have demonstrated that partial exocytotic release is prevailing in neuroendocrine cell models. However, due to complicated structure and culture process, few studies have been carried out using neurons, especially human neurons. Here, dopamine (DA) release from individual vesicles and DA content stored within vesicles were quantified from induced pluripotent stem cell-derived DA neurons with electrochemical techniques. The results indicate that around 61% of the total vesicular DA content is released from these neurons during exocytosis. The vesicular content quantified in DA neurons is significantly higher than that in undifferentiated neural progenitor cells, owing to the increased appearance of dense-core vesicles that are able to store more DA molecules than the clear vesicles. When the neurons are differentiated with BAY-K8644, which stimulates neuronal maturation as well as DA release, the release fraction rises to 91%. The use of BAY-K8644 can be considered as chronic stimulation and leads to similar effects on exocytosis as repetitive stimulation, which triggers short-term plasticity. This study demonstrates partial release in DA transmission in human neurons and provides a link between neuronal maturation and the formation of plasticity. Furthermore, this work suggests that the fraction of release in exocytosis at human neurons may be a factor in determining plasticity.

## Introduction

Neuronal communication is typically carried out by secretory vesicles *via* a process called exocytosis. To be prepared for exocytosis, vesicles that are releasable at once need to be docked and then primed to the cell membrane. When an electrical signal or a chemical stimulant reaches the release site, intracellular calcium level is elevated, either *via* the influx of calcium ions through calcium channels, or the release of intracellular calcium storage, which triggers the formation of the fusion pore between the vesicle and the cell membrane. This allows the vesicular neurotransmitters to be released to the extracellular space (Miledi, 1973). Exocytosis has long been considered as an all-or-none process (Del Castillo and Katz, 1954). However, during the last decade, emerging research has demonstrated the prevalence of partial release (Wu et al., 2019; Wang and Ewing, 2020; Yang et al., 2021; Nguyen et al., 2022; Borges et al., 2023; Gu et al., 2024). Notably, it shows the possibility to regulate neurotransmission at the level of single vesicles and its significance, for example, in activity-dependent plasticity (Gu et al., 2019; Zhang et al., 2024).

Neural progenitor cells (NPCs) are derived from induced pluripotent stem cells (iPSCs) and can be subsequently differentiated into certain types of neuronal cells and glia. Here, NPCs were differentiated into dopamine (DA) neurons to study neuronal DA transmission. DA, a neurotransmitter as well as a neuromodulator, has several essential pathways in the brain, including the reward system and the motor control (Berridge, 2007). In addition to that, the involvement of the DA system in aging, disorders, and diseases has been widely studied and acknowledged (Wong et al., 1984; Wang et al., 1998; Ceravolo et al., 2010; Dobryakova et al., 2015). Therefore, understanding the fundamentals of DA signaling is of great importance, especially in neuronal cells. Exocytosis of DA molecules has been shown to be partial in pheochromocytoma (PC12) cells, followed by several studies attempting to understand how partial release is regulated by different factors (Borges et al., 2023). However, most studies were carried out in neuroendocrine cell models. Although the machinery of exocytosis is conserved among multicellular organisms, morphology of the neuron is highly complex. Moreover, neurons possess multiple release sites and distinct secretory vesicle pools compared to other cell models (Fujise et al., 2025). Studying exocytosis from DA neurons will expand our knowledge regarding DA signaling as well as partial exocytotic release.

Two electrochemical techniques, single-cell amperometry (SCA) and intracellular vesicle impact electrochemical cytometry (IVIEC) were applied in this work to monitor DA transmission in neurons. Both techniques are capable of providing sufficient temporal resolution, down to the millisecond timescale, to resolve individual vesicular events. SCA was developed in the early 1990s to quantify the number of molecules released during exocytosis from single cells (Leszczyszyn et al., 1990; Wightman et al., 1991). By placing a micro- or nanoelectrode above a single cell, electroactive neurotransmitters (*e.g.*, DA) that are released from single vesicles in response to a stimulus are oxidized on the electrode surface and detected as amperometric spikes. Quantitative and dynamic information about an exocytotic release event can be obtained from the area and the shape of the spike, respectively (Schroeder et al., 1996). IVIEC was introduced in 2015 for quantifying the number of molecules stored inside vesicles within single cells (Li et al., 2015). This is typically achieved by piercing the sharp tip of a flame-etched nanoelectrode into the cytoplasm, allowing the neurotransmitter content of vesicles to be oxidized and measured on the tip (Li et al., 2018). Other types of electrodes can also be utilized to conduct IVIEC, for example, open carbon nanopipettes (CNPs) that have been used to measure vesicles with different size ranges intracellularly (Hu et al., 2022). Transmission electron microscopy (TEM) offers ultrastructural images and is used to visualize subcellular structures such as vesicles. Clear vesicles (CVs) are observed as transparent compartments under TEM, whereas dense-core vesicles (DCVs) are seen to be darker due to the electron-dense property of the protein core. TEM was applied here as a complementary technique to assist our understanding of the electrochemical results.

In this paper, a FOXA2-ntdTomato reporter line was established and by combining SCA and IVIEC, DA release from single differentiated FOXA2 DA neurons was studied. The results show that on average, these neurons release approximately 61% of their vesicular DA content during exocytosis. Compared to vesicular content in undifferentiated NPCs, vesicles in DA neurons store significantly larger amount of DA molecules. This can be explained by the clusters of DCVs that are present in DA neurons under TEM, whereas NPCs have mostly CVs. BAY-K8644 is able to induce neuronal maturation and stimulate DA release. We found that differentiation with BAY-K8644 (chronic stimulation) leads to similar effects on both exocytosis and intracellular calcium dynamics as repetitive stimulation (acute stimulation), which triggers short-term plasticity. These together demonstrate the complex nature of DA transmission in human neurons and suggest that plasticity in DA signaling can occur on the vesicular level and may be related to neuronal maturation changing the fraction of transmitter cargo released.

## Results

### Establishment and characterization of the human midbrain FOXA2 DA neurons

The FOXA2-ntdTomato reporter iPSC line was generated by cotransfecting control DA NPCs with a CRISPR-Cas9 vector containing a gRNA targeting the 3′ end of FOXA2 (Figure 1a), along with the donor plasmid pUC19-FOXA2-T2A-2xNLS-tdTomato-F2A-Puro (Figure 1b). The tdTomato-positive cells, as shown in Figure 1c, were then isolated and reprogrammed to iPSCs, which were subsequently selected and expanded.

To confirm the neuronal and DA characteristics of the cell line upon differentiation, fluorescent imaging was carried out using confocal microscopy to examine the expression level of FOXA2 inside the cell body within neuronal cultures, which have been differentiated for 2 weeks. As shown in Supplementary Figures S1A, a majority of cells possess a visible level of FOXA2 signal. Immunostaining against beta 3 tubulin, a microtubule element found mostly in neurons (Caccamo et al., 1989), and tyrosine hydroxylase, the essential enzyme for DA synthesis, further confirmed the characteristics of DA neurons (Supplementary Figures S1B and S1C).

### Dopamine release and storage quantified from single differentiated DA neurons show partial release

Differentiated DA neurons were identified by the presence of neurites around the cell body as well as multiple branches (Figure 2a left panel). Exocytotic release was measured by SCA with electrodes that have conical-shaped tips and aimed at a few areas of the neurons, including boutons, bulb-looking structures along the axons where release typically occurs, and/or intersections between axons (Figure 2a right panel). Exocytosis was stimulated chemically with a solution containing 100 mM K$^+$ for a continuous 10 s. As DA is electroactive, its release can be detected on the electrode surface *via* electrochemical oxidation. Release events from single vesicles were observed as individual amperometric spikes (Supplementary Figure S2A) and the area under the spike was integrated to calculate the number of DA molecules (see Materials and Methods in Supplementary Materials for detailed calculation) (Leszczyszyn et al., 1990). As shown in Figure 2b, the average release per neuron was quantified to be 111000 ± 14000 molecules (data from each neuron can be found in Supplementary Figure S2B). We did not measure exocytotic release from undifferentiated NPCs as the release sites, mature axons, and boutons were not clearly formed.

IVIEC was applied with CNPs (Hu et al., 2022) to measure the number of DA molecules stored originally within individual vesicles. The use of CNPs enables an easier penetration through the neuronal membrane, which is more flat compared to typical cell models used to study neurosecretion, e.g., chromaffin cells. The diameter of the pore of the CNPs is 1–1.5 μm, which covers the sizes of both CVs and DCVs and, thereby, is capable of detecting all-sized vesicles. IVIEC from undifferentiated NPCs was first measured by piercing CNPs into the cell body area. Vesicles then diffused into the open tip of the CNPs and the content was detected on the carbon surface coated inside of the CNP electrodes. Similar to SCA, DA molecules stored within single vesicles gave rise to amperometric spikes, which were subsequently integrated to obtain the number of molecules. We measured IVIEC from NPCs here to examine how vesicular DA content is affected by differentiation. Vesicles in NPCs store a mean level of 87000 ± 19000 DA molecules, as depicted in Figure 1b (data from each neuron can be found in Supplementary Figure S2B). After differentiation for 2 weeks, DA neurons possess significantly higher amounts of vesicular neurotransmitter content, with the average being 182000 ± 18000 molecules (Figure 1b, and data from each neuron can be found in Supplementary Figure S2B). IVIEC from DA neurons was measured from either cell bodies or boutons to give better comparisons to both IVIEC from NPCs and SCA from DA neurons (no significant differences in the data and number of detected spikes were observed for IVIEC measurements from different cell areas). When comparing between the SCA and IVIEC obtained from the same type of neuron, it was noted that these vesicles store many more DA molecules than what is being released during exocytosis, with the fraction (ratio between molecules from SCA and from IVIEC) being 0.61. This result shows that vesicles in human DA neurons release only a part of the neurotransmitter content during exocytosis under normal conditions,

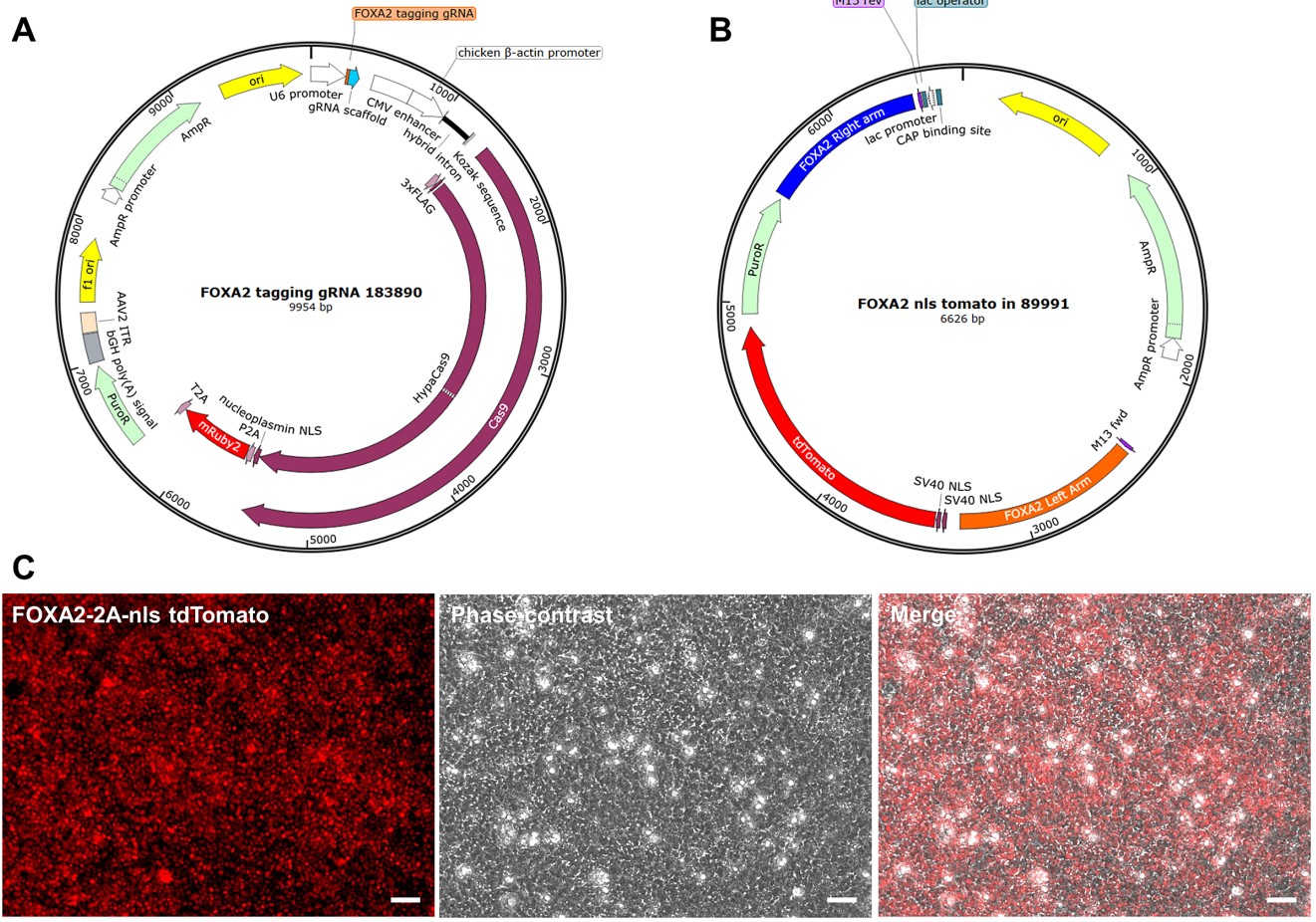

**Figure 1.** Generation of a FOXA2-ntdTomato Reporter iPSC Line. (a) FOXA2-targeting CRISPR-Cas9 plasmid map. The guide RNA targeting the FOXA2 stop codon was designed using CRISPick, synthesized by GenScript and cloned into the pX459-HypaCas9-mR2-AAVS1_sgRNA vector from Addgene. (b) Donor plasmid map. The donor vector was generated by inserting the left and right homology arms flanking the FOXA2 stop codon into the pUC19-FOXA2-T2A-2xNLS-tdTomato-F2A-Puro backbone using SbfI, NheI, AscI, and NotI restriction sites. (c) Live-cell imaging of FOXA2-ntdTomato reporter expression. Representative fluorescent and phase-contrast images of DA NPCs showing nuclear tdTomato fluorescence in FOXA2-positive cells. Images were acquired at 20x magnification. Scale bars: 50 μm.

in agreement with what has previously been reported for other types of neurons (Larsson et al., 2020; Yang et al., 2021; Gu et al., 2024).

### *BAY-K8644 enhances both DA release and release fraction as a stimulant for neuronal maturation, but inhibits calcium influx*

As plasticity in neurotransmission can be induced *via* stimulation (Gu et al., 2019), we further investigated how DA transmission is regulated when NPCs are differentiated for 2 weeks in the presence of a chemical stimulant, BAY-K8644. In addition to being a calcium channel agonist, BAY-K8644 itself has been suggested to induce the release of DA (Liu et al., 2007). Moreover, it elevates DA levels in differentiated neurons and shows the possibility to promote neuronal maturation during differentiation and was therefore selected for this study (Jefri et al., 2020). Results from SCA indicate a significant increase of the average number of DA molecules released per neuron ($167000 \pm 13000$ molecules, Figure 2b, and data from each neuron can be found in Supplementary Figure S2B) compared to the ones differentiated without BAY-K8644, while the average amount of DA content within vesicles remains nearly unaltered ($183000 \pm 18000$ molecules). The release fraction calculated for the neurons differentiated with BAY-K8644 is 0.91.

During exocytosis, the formation of a fusion pore between the vesicle and the cell membrane allows the efflux of neurotransmitters to the extracellular volume. The activity and the time course of the fusion pore can be reflected by certain parameters of the amperometric spike, including $t_{1/2}$, $t_{rise}$, and $t_{fall}$, (Supplementary Figure S2A), which correlate with the duration, the opening, and the closing of the fusion pore, respectively. On average, release events measured from DA neurons differentiated in the presence of BAY-K8644 exhibited longer duration (Supplementary Figure S2C), which leads to more DA molecules being released. In general, the duration of the exocytotic release events measured from these neurons ($t_{1/2}$ about 0.7 ms) is shorter than that observed from neuroendocrine cell models (e.g., $t_{1/2}$ from PC12 cells is larger than 1 ms and $t_{1/2}$ from adrenal chromaffin cells is even larger, a few milliseconds).

The opening of calcium channels and the subsequent influx of calcium ions are essential for the initiation of the fusion process during exocytosis. As BAY-K8644 is a calcium channel agonist, we investigated how calcium influx is influenced by chronic BAY-K8644 treatment when DA neurons are stimulated for exocytosis. As indicated in Table 1, DA neurons were stimulated with the 100 mM $K^+$ stimulation solution about 15 s after the beginning of the recording. An abrupt elevation of intracellular calcium signal was observed. The fluctuations of calcium signal intensity relative

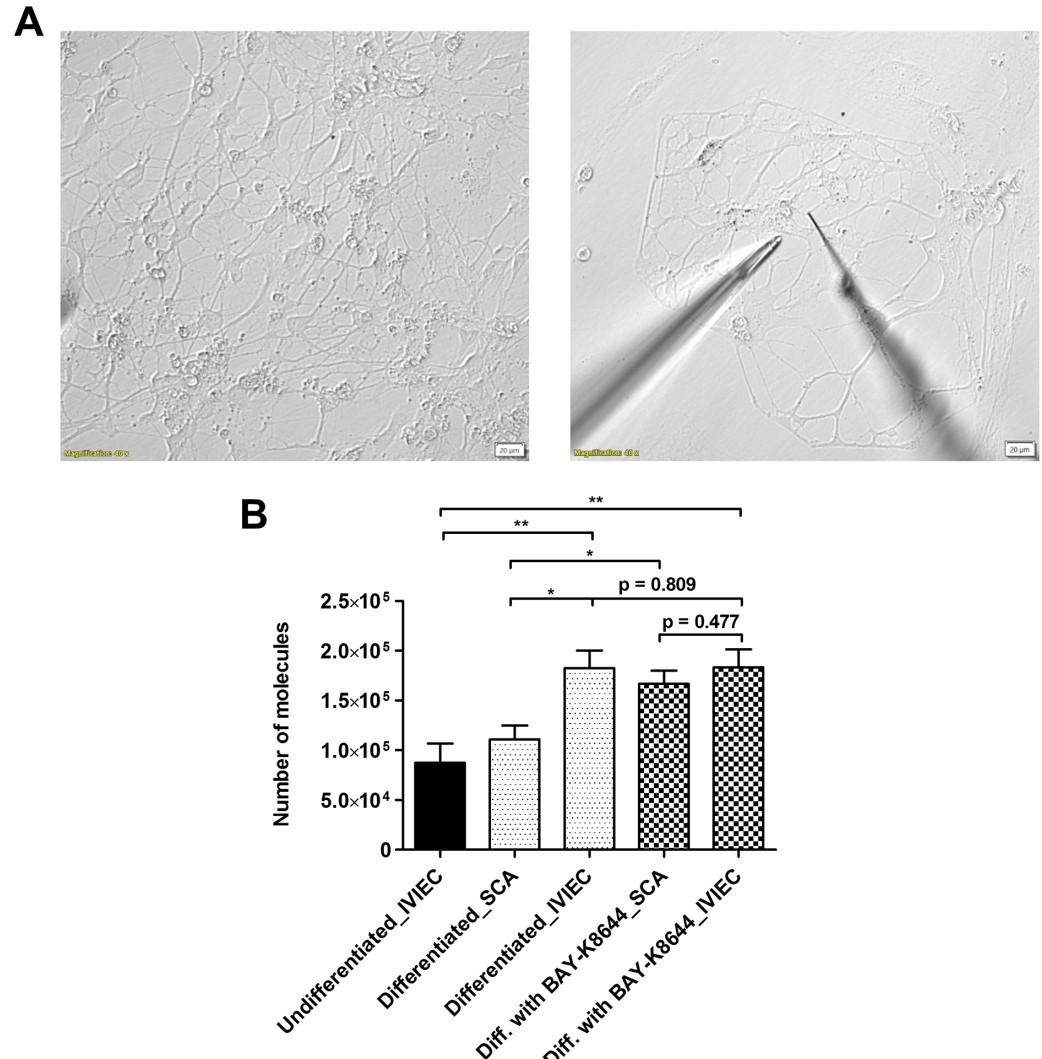

**Figure 2.** (a) Representative bright field images showing the morphology of differentiated FOXA2 DA neurons (left panel), and positions of an electrode (on the right) and a stimulation pipette (on the left) within the neuronal network to study DA release (right panel). (b) Comparison of average numbers of DA molecules measured by SCA and IVIEC under different conditions, including NPCs (undifferentiated), differentiated DA neurons, and DA neurons differentiated with BAY-K8644. DA release and vesicular DA content were quantified by SCA and IVIEC, respectively. In total, 7 cells were counted for IVIEC from NPCs and over 10 cells were counted for the rest of the groups. Error bars represent means of medians ± SEM. Data sets were compared with Mann–Whitney test, *p < 0.05, **p < 0.01, and other p values are indicated in the graph.

to the baseline level during the first 15 s were calculated and it was found that differentiation in the presence of BAY-K8644 induces a level of calcium influx in response to stimulation lower than that of the DA neurons differentiated without BAY-K8644.

Overall, the data above indicate that chronic BAY-K8644 treatment during neuronal differentiation is able to trigger increased release of DA molecules during exocytosis as well as elevated release fraction, whereas the amount of calcium influx required for fusion is lower than for the neurons without BAY-K8644 treatment, which is a possible outcome of the stimulation effect of BAY-K8644.

### *Increased amount of DCVs due to neuronal maturation correlates with elevated vesicular DA content*

To understand the possible reason governing the significant increase of vesicular DA content quantified by IVIEC upon differentiation, TEM was performed on both NPCs and differentiated DA neurons to compare the morphology of vesicles. Figure 3a shows an example of the cell body area of the NPCs, which was occupied by clusters of CVs

as well as other organelles. The presence of DCVs can be observed frequently in differentiated DA neurons (Figure 3b), meanwhile clusters of CVs can still be found (Figure 3c).

We further pooled out all events measured by IVIEC and examined the distributions of the number of molecules under all three conditions. Results of IVIEC from NPCs are depicted as normalized frequency histogram of number of molecules, as well as log distribution of number of molecules, in Supplementary Figures S3A and 4A, respectively. A small shoulder on the right is observable in Supplementary Figure S3A, and the log distribution was fitted into two Gaussian distributions with the means of the first and the second distribution being 38000 and 121000 molecules, respectively. Interestingly, upon differentiation, molecule distribution from DA neurons exhibits only single Gaussian distribution, which can be seen in Supplementary Figures S3B and 4B. The mean value of the distribution is 124000 molecules, which resembles the mean of the second distribution measured from NPCs. These together suggest that the increased presence of DCVs observed under TEM upon differentiation leads to an elevated vesicular DA content quantified

**Table 1.** Results from calcium imaging showing the dynamic changes of fluo-4 fluorescence intensity during stimulated exocytosis from differentiated DA neurons without or with BAY-K8644

| Time of recording | 0–15 s | 15–30 s | 30–45 s | 45–60 s | 60–90 s |
| --- | --- | --- | --- | --- | --- |
| Average fluo–4 fluorescence intensity from differentiated neurons (n = 9) | 4274 | 6941 | 6918 | 6968 | 6350 |
| Variation relative to baseline level (0–15 s) | 0 | +62.4% | +61.9% | +63.0% | +48.6% |
| Average fluo–4 fluorescence intensity from differentiated neurons with BAY-K8644 (n = 18) | 3518 | 5087 | 5148 | 5058 | 4746 |
| Variation relative to baseline level (0–15 s) | 0 | +44.6% | +46.3% | +43.8% | +34.9% |

*Notes:* Cells were incubated with 1 μM fluo-4 calcium indicator for 30 min prior to the calcium imaging experiments. Cells were stimulated for exocytosis 15 s after the beginning of the recording with 100 mM$K^+$ stimulation solution and intracellular calcium level was recorded for 90 s. The results were divided into time intervals of 15 s for the first 60 s. The average fluorescence intensity from each cell for each time interval was first calculated, and then the average from all cells for each time interval was calculated and shown in the table. Average fluorescence intensity during 0–15 s was used as the baseline level and relative alterations of fluorescence intensity after the stimulation were calculated as percentages and indicated in the table.

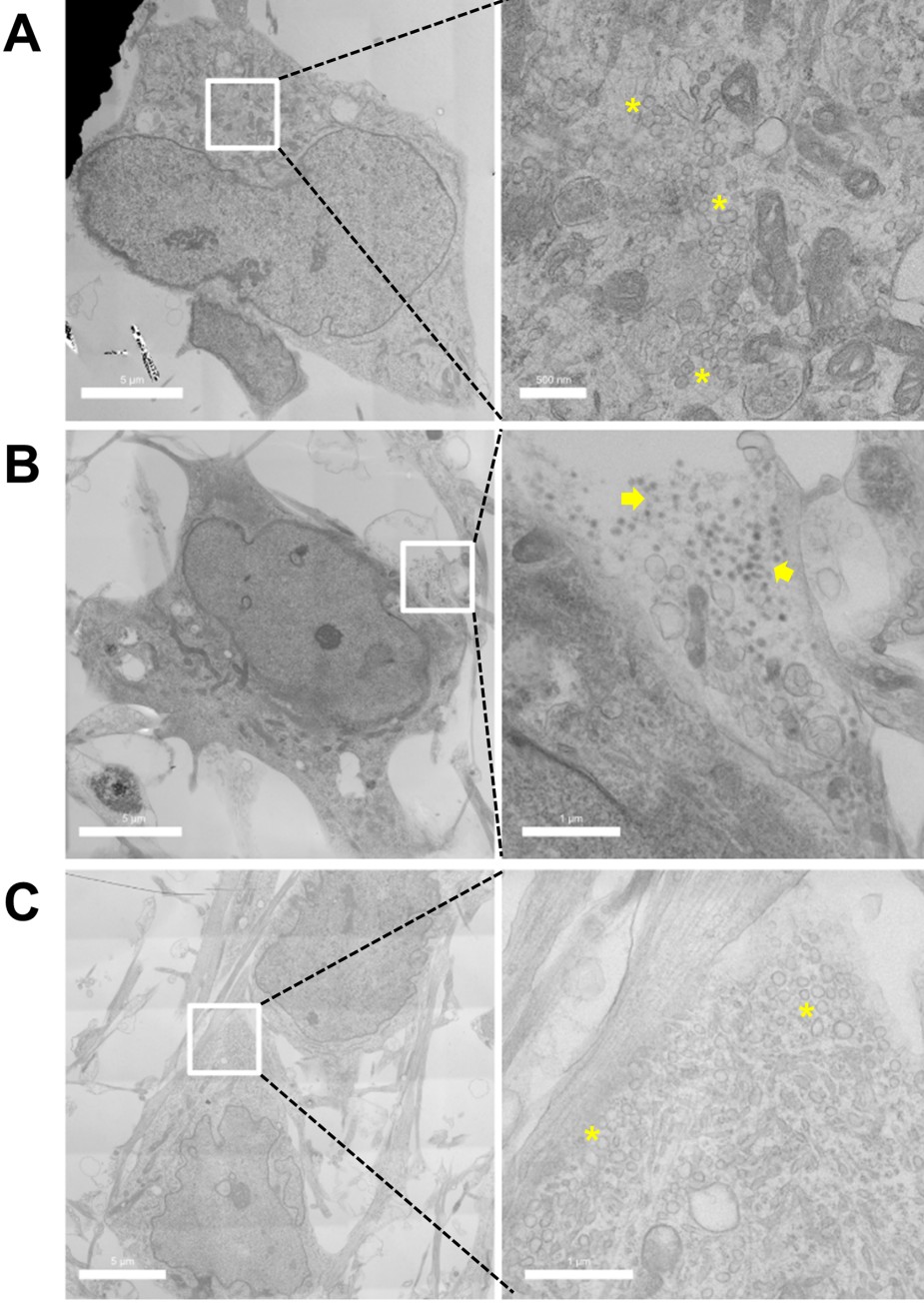

**Figure 3.** TEM images of (a) NPCs and (b–c) differentiated DA neurons showing the presence of CVs and DCVs. Left panels show overviews of cells and on the right side are the magnifications of the cropped areas. Asterisks indicate clusters of CVs and groups of DCVs are indicated by arrows.

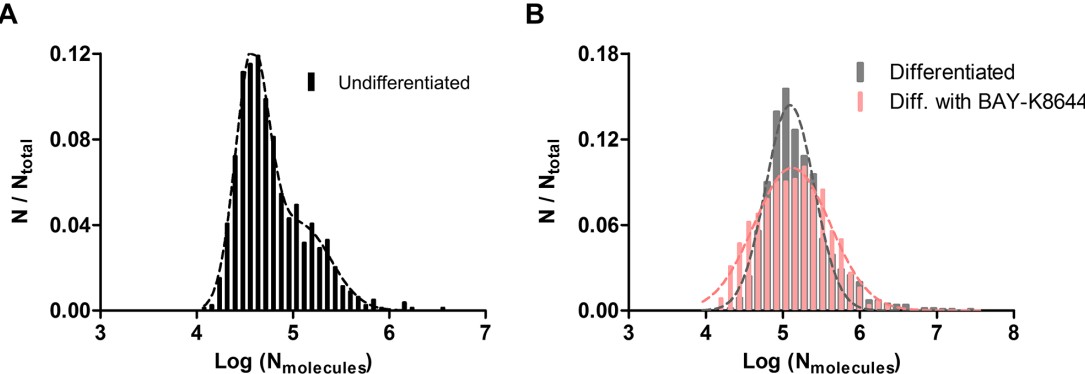

**Figure 4.** (a) Log distribution of number of molecules quantified by IVIEC from NPCs (undifferentiated, 790 events from 7 cells), bin size = 0.08. Data were fitted into two Gaussian distributions. (b) Log distributions of number of molecules quantified by IVIEC from differentiated DA neurons without (1256 events from 25 cells) or with BAY-K8644 (1063 events from 18 cells), bin size = 0.12. Both groups were fitted into single Gaussian distributions.

by IVIEC. When DA neurons are differentiated with BAY-K8644, the distribution of molecular count appears to be a single Gaussian, but this distribution is slightly wider relative to cells differentiated without BAY-K8644 (Figure 4b), with the mean of the distribution being 133000 molecules. Molecule count distributions from all three groups were also combined to give a better comparison and can be found in Figure S3C in the Supplementary Materials.

## Discussion

Partial release has previously been reported in different cell types as well as in living neurons (Li et al., 2015; Wu et al., 2019; Larsson et al., 2020; Hatamie et al., 2021; Wang et al., 2021; Yang et al., 2021; Nguyen et al., 2022; Borges et al., 2023; Gu et al., 2024). Notably, the release fraction varies greatly among cell types and can even be largely differential within the same set of vesicles (Gu et al., 2024). In addition, drug treatment or cellular activity has been demonstrated to alter release fraction (Gu et al., 2019; Wang and Ewing, 2020). Here, we found that mean release fraction from differentiated human DA neurons was 0.61. This fraction is similar to what has been quantified previously from PC12 cells, which store and release mainly DA molecules from their DCVs (Greene and Tischler, 1976; Li et al., 2015). However, not all vesicles within a cell undergo exocytosis in response to a stimulus and for PC12 cells as an example, only about 20% of vesicles are releasable at once (Gu and Ewing, 2021). Moreover, the releasable pool of vesicles in PC12 cells may be heterogeneous (Westerink et al., 2000), adding complexity to the nature of exocytosis as well as to the calculation of the release fraction.

As for the differentiated DA neurons studied in this work, we observed different types of vesicles (CVs and DCVs) under TEM. The observation of multiple pools of vesicles with distinct morphology in iPSC-derived DA neurons has previously been reported (Fujise et al., 2025). DCVs typically contain a protein core that consists of chromogranins (Cgs) (Huttner et al., 1991). The strong binding affinity of Cgs toward neurotransmitters like DA makes it possible for DCVs to accumulate a relatively high concentration of neurotransmitters (Videen et al., 1992; Albillos et al., 1997), especially comparing to CVs. Therefore, the high number of DCVs in the DA neurons likely accounts for the increased number of molecules quantified by IVIEC relative to the value obtained from the NPCs. The distribution of molecule count in NPC vesicles can be fitted into two Gaussian distributions, with the mean value of the second

distribution being nearly the same as the mean value from the DA neurons. This indicates that the NPCs possess mostly CVs but also a small fraction of DCVs, whereas upon differentiation and neuronal maturation, a majority of vesicles are DCVs and the vesicular DA storage appears to be relatively homogeneous within the vesicle population. Although both types of vesicles are able to release, the capability of release may differ under specific conditions (Borisovska et al., 2013; Zhang et al., 2024). As neuronal maturation leads to an increased amount of DCVs, it is possible that these vesicles are more mature and are, thereby, more prone to release than the CVs. Considering that all types of DA vesicles are detectable by IVIEC (CVs and DCVs including releasable and reserved ones) while exocytosis is measured from mature and immediate releasable vesicles, the actual release fraction can be lower than 0.61.

Differentiation in the presence of BAY-K8644 elevates both the average number of DA molecules released and the release fraction without affecting the average vesicular DA content significantly, when compared to the DA neurons differentiated without BAY-K8644. It has been reported that BAY-K8644 stimulates the process of neuronal maturation and moreover, it stimulates DA release (Liu et al., 2007; Jefri et al., 2020). The effect of BAY-K8644 in this work is similar to chronic stimulation. Acute stimulation on the other hand, which is linked to the initiation of short-term plasticity, has been studied previously both *in vitro* and *in vivo via* repetitive stimulation with relatively short time intervals (Gu et al., 2019; Somayaji et al., 2020; Zhang et al., 2024). In PC12 cells, repetitive stimulation leads to an extended fusion process, more molecules being released during exocytosis, higher release fraction, but lower vesicular content on average (Gu et al., 2019). The same trend in exocytosis was observed from DA neurons isolated from rats (Zhang et al., 2024). Here, we demonstrate that BAY-K8644 treatment induces effects similar to acute stimulation, except that the average vesicular DA content remains nearly unaltered. However, the molecule distribution pooled from all IVIEC events appears to be slightly wider, which can be explained by the dual effects caused by BAY-K8644 incubation. The chronic stimulation effect induced by BAY-K8644 triggers constant DA release, which depletes neurotransmitter content from a fraction of vesicles and is represented by the left shoulder in the distribution. In addition, BAY-K8644 can induce neuronal maturation, which results in an increased number of mature vesicles storing higher amount of DA molecules. This explains the right shoulder observed in the distribution. In general, we found that the effects of chronic stimulation on neurotransmission resemble what has been reported for acute stimulation, and

these further support the possible connection between fraction of neurotransmitter release and plasticity (Gu et al., 2019).

Calcium imaging was carried out in this study to examine the alteration of intracellular calcium level when triggering exocytosis with or without BAY-K8644 incubation during differentiation. Acute application of BAY-K8644 was shown to induce significant influx of calcium ions and enhance firing frequency in DA neurons (Jefri et al., 2020). Here, BAY-K8644 was applied chronically and we observed less calcium influx relative to the neurons differentiated without BAY-K8644. As the elevation of intracellular calcium level is essential for the initiation as well as the frequency of exocytosis (Heidelberger et al., 1994; Finnegan and Wightman, 1995), it is possible that the chronic stimulation effect of BAY-K8644 makes the neurons adapted to bear fewer calcium channels and meanwhile, more sensitive to the amount of calcium influx. Thereby, less calcium ions are needed to trigger exocytosis. In PC12 cells, different sizes of DCVs express different isoforms of synaptotagmin, the calcium sensor. Large DCVs bear more synaptotagmin VII, which exhibits higher sensitivity toward calcium compared to the other isoforms (Zhang et al., 2011). As BAY-K8644 incubation possibly induces the maturation of vesicles, this can lead to increased calcium sensitivity. Interestingly, a reduced intracellular calcium level was also observed in cells upon acute stimulation (Gu et al., 2019), a phenomenon related to plasticity.

We speculate that the impact of this work on plasticity in human neurons is that it provides an alternative or supplementary mechanism for developing plasticity eventually leading to learned synaptic responses. We propose a model whereby the fraction of transmitter release is important in developing plastic responses in these neurons. We speculate that the higher fraction released is equivalent to increasing the frequency of events in terms of altering synaptic concentration of transmitter. A change in SNARE complex bringing about a 50% change in exocytosis event frequency will increase the synaptic transmitter to the same extent as changing the fraction released from 60% to 90%.

It is very likely that factors regulating presynaptic plasticity involve a complex interaction of the SNARE complex affecting release event frequency and fusion pore dynamics affecting fraction release. It is possible these processes augment each other as the effects will be additive. The result is plasticity with nuance for variation and different plastic responses depending on the effector. This model points to both protein and membrane structure that can be altered to affect plasticity, leading to a broader range of solutions to aiding learning and memory as well as treating neurodegenerative diseases. The rise in release fraction to 91% following cell maturation with BAY-K8644 might be indicative of the system compensating for loss of learning with aging or neurodegenerative disease.

## Conclusions

To summarize, DA release and vesicular DA content were quantified from human differentiated FOXA2 DA neurons with electrochemistry. We found that mean release fraction from these DA neurons was 0.61. The release is clearly partial, but the actual fraction might be smaller due to the complexity of subpopulations of the vesicles. TEM imaging reveals that NPCs possess mainly CVs, whereas DA neurons have both CVs and DCVs, which contribute to the higher amount of vesicular content measured upon differentiation and neuronal maturation. Differentiation with BAY-K8644, a neuronal network enhancer and cell stimulant, can be considered chronic stimulation and leads to effects that are similar to acute stimulation, which is triggered by repetitive stimulation and induces short-term plasticity. This study shows the complex nature of DA signaling in human neurons and helps to further understand neuronal maturation and plasticity. Finally, we offer a hypothesis for a role of partial release in presynaptic plasticity and aging.

## Materials and Methods

The methods used in this study included plasmid constructs and generation of FOXA2-ntdTomato reporter iPSC line, electrochemical measurements, calcium imaging, TEM imaging, and confocal imaging. Detailed description of materials and methods is provided in the Supplementary Materials.

**Open peer review.** To view the open peer review materials for this article, please visit http://doi.org/10.1017/qrd.2026.10019.

**Supplementary material.** The supplementary material for this article can be found at http://doi.org/10.1017/qrd.2026.10019.

**Acknowledgments.** We thank Anna Larsson, Pieter Oomen, and Ying Wang for their contributions at the early stage of this project. We also acknowledge the Centre for Cellular Imaging at the University of Gothenburg and the National Microscopy Infrastructure, NMI (VR-RFI 2019-00217) for providing assistance in calcium imaging and TEM imaging.

**Author contribution.** A.G.E., C.G., A.A.L., and S.M. designed the study; H.P., A.N., and C.E. established and provided the cell line; C.G., A.A.L., S.M., and S.R. performed the research; C.G. analyzed the data and wrote the first draft; A.G.E. reviewed and edited the manuscript; A.G.E. and C.E. acquired the funding. All authors have given approval to the final version of the manuscript.

**Financial support.** The Andrew G. Ewing lab was funded by the European Research Council (ERC Advanced Grant, project number 101199467), the Knut and Alice Wallenberg Foundation, and the Swedish Research Council (VR, project number 2022–03523). The Carl Ernst lab was funded by the Djavad Mowafaghian Foundation.

**Competing interests.** The authors declare no conflict of interest.

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
