## [Reviewer Report]

This is an interesting article in which Ewing and coworkers have investigated the release of dopamine molecules from dopaminergic neurons using electrochemical methods. These authors have been at the forefront of studying dopamine exocytosis from single neuron model cells and neurons using highly-sensitive and highly-resolving electrochemical, mass spectrometry, and various microscopic tools and methods. In this study, they set out to study the percentage of released dopamine from a new cell type, human iPSC-Derived FOXA2 dopaminergic neurons.

The particular electroanalytical methods they chose are single-cell amperometry (SCA) and intracellular vesicle impact electrochemical cytometry (IVIEC), both of which allow single vesicles to be analyzed with millisecond and sub-millisecond time resolution. By measuring the amount of released dopamine in each exocytosis event and the amount of dopamine molecules from each intracellular vesicles detected with IVIEC, they were able to quantitatively show 61% of the vesicular dopamine molecules are being released during stimulated exocytosis from single differentiated FOXA2 neurons. Moreover, their results also show that vesicles in differentiated neurons contain more dopamine molecules compared to those undifferentiated neural progenitor cells.

Besides their electrochemical data, the authors also provided imaging results using transmission electron microscopy, which clearly show the presence of more dense core vesicles in differentiated dopamine neurons, which are responsible for the higher dopamine content being detected in their electrochemical recording.

The present paper is a great demonstration of the usefulness of single-cell amperometry and intracellular vesicle impact electrochemical cytometry for single cell analysis. The paper is clearly written with detailed information on how different methods were executed and how results were analyzed. I am happy to suggest its publication in QRBD.

Line 307: “provides and alternative...” is likely “provides an alternative”

Ref 4 seems to be missing some information.

---

## [Reviewer Report]

This paper provides vital insights into the exocytosis of dopamine from neurons, which as highlighted by the authors has barely been conducted. Therefore this provided new and novel findings. The following aspects are suggested to enhance the manuscript

1. When differentiated, it is mentioned that measurements were conducted in a few areas of the neurons but were there any differences observed if measurements were conducted in cell body, axonal or intersections.

2. The authors have previously measured other characteristics of the individual events to provide insight into the vesicular release dynamics. Did these alter and did they show features that were different that typical exocytotic events.